# Congenital Microcephaly: A Debate on Diagnostic Challenges and Etiological Paradigm of the Shift from Isolated/Non-Syndromic to Syndromic Microcephaly

**DOI:** 10.3390/cells12040642

**Published:** 2023-02-16

**Authors:** Maria Asif, Uzma Abdullah, Peter Nürnberg, Sigrid Tinschert, Muhammad Sajid Hussain

**Affiliations:** 1Cologne Center for Genomics (CCG), Faculty of Medicine, University Hospital Cologne, University of Cologne, 50931 Cologne, Germany; 2Center for Molecular Medicine Cologne (CMMC), Faculty of Medicine, University Hospital Cologne, University of Cologne, 50931 Cologne, Germany; 3University Institute of Biochemistry and Biotechnology (UIBB), PMAS-Arid Agriculture University, Rawalpindi, Rawalpindi 46300, Pakistan; 4Zentrum Medizinische Genetik, Medizinische Universität, 6020 Innsbruck, Austria

**Keywords:** congenital microcephaly (CM), MCPH, microcephaly primary hereditary, isolat-ed/non-syndromic, syndromic microcephaly, genetic modifiers, brain-specific splicing events, multi-organ phenotype of CM

## Abstract

Congenital microcephaly (CM) exhibits broad clinical and genetic heterogeneity and is thus categorized into several subtypes. However, the recent bloom of disease–gene discoveries has revealed more overlaps than differences in the underlying genetic architecture for these clinical sub-categories, complicating the differential diagnosis. Moreover, the mechanism of the paradigm shift from a brain-restricted to a multi-organ phenotype is only vaguely understood. This review article highlights the critical factors considered while defining CM subtypes. It also presents possible arguments on long-standing questions of the brain-specific nature of CM caused by a dysfunction of the ubiquitously expressed proteins. We argue that brain-specific splicing events and organ-restricted protein expression may contribute in part to disparate clinical manifestations. We also highlight the role of genetic modifiers and de novo variants in the multi-organ phenotype of CM and emphasize their consideration in molecular characterization. This review thus attempts to expand our understanding of the phenotypic and etiological variability in CM and invites the development of more comprehensive guidelines.

## 1. The Definition of Congenital (Primary) Microcephaly and Its Subtypes

Microcephaly opens an aperture to glimpse into the evolution of the human brain, thus uncovering the veils of its architectural orchestration and immense adaptability. Microcephaly is defined as a reduction in the occipital–frontal head circumference (OFC) of at least 2 SD below the mean of age, gender, and ethnicity-matched populations [1]. It underlies the reduced volume of the cerebral cortex, and thus the brain [2]. “Primary” microcephaly means an abnormal OFC at birth [1] and is also known as “congenital microcephaly” (CM) [3]. CM is the currently favored nomenclature and is broadly divided into three categories: (1) isolated microcephaly, that is, without any further clinical (cerebral or extra-cerebral) abnormalities, including no developmental disability or intellectual disability; (2) non-syndromic microcephaly with neurological or psychiatric features, but without cerebral malformations or extra-cerebral abnormalities; and (3) syndromic microcephaly with malformations of the cerebrum and/or extra-cerebral morphologic or functional anomalies. The term microcephaly primary hereditary (MCPH) refers to the genetic forms of CM [4,5]. Adopting these definitions is often difficult and has therefore not been handled uniformly.

## 2. The Inconsistent Diagnostic Criterion

Clinically, the first step towards the diagnosis of CM is measuring the OFC for a significant reduction. However, different cutoff values are used to define microcephaly. For instance, the American Academy of Neurology and the Practice Committee of the Child Neurology Society have recommended a cutoff of <−2 SD OFC at the time of birth, while <−3 SD would indicate a severe degree of microcephaly [6]. However, multiple studies followed −3 SD as a baseline [5].

The reference values being used to calculate the deviation also vary considerably. As already highlighted by van der Hagen et al. [7], for most of the non-European patients, the SD of OFC is often calculated with reference to values taken from studies performed on Nellhaus (Caucasian children) [8] or Prader (Swiss children) [9], irrespective of the ethnicity of the patients. It should be noted that the mean OFC of children from industrial countries is larger than that shown by the WHO standard values (www.who.int/childgrowth/en accessed on 13 February 2023) [7]. Therefore, it is of critical importance that population-specific data on OFC are taken into consideration when defining microcephaly; the existing data on head circumferences from different populations of Swedish (265,456 singleton neonates), Norwegian (77,044 families), Finnish (533,666 singletons and 15,033 twins) and European, Chinese and South Asian ancestries (2695 infants) may be taken into account for patients of the respective population [10,11,12]. Similarly, the growth charts published by the Centers for Disease Control and Prevention (CDC; www.cdc.gov/growthcharts/ accessed on 13 February 2023) may also be utilized for post-infantile patients of American ancestry.

Furthermore, measuring parental OFCs should be considered for the diagnosis of familial microcephaly. Additionally, a pattern in SD values of OFC of patients with mutations in the same gene could be taken into consideration for a more precise genotype–phenotype correlation.

### Inconsistencies with Using the Terms Primary Microcephaly and MCPH

Hereditary CM (MCPH) is sub-categorized—depending on the absence or presence of co-existing features—into isolated, non-syndromic, and syndromic forms. It is important to emphasize that the term “MCPH” is often used for autosomal recessive non-syndromic microcephaly only (see MIM #251200, MCPH1), which is characterized by microcephaly and intellectual deficiency; in some patients, short stature was present. Non-syndromic microcephaly almost exclusively follows an autosomal recessive mode of inheritance and is known to be associated with 30 different genes [13]. However, mutations in one of the MCPH genes, *WDFY3*, act in an autosomal dominant manner and are associated with MCPH18, characterized by microcephaly with mild to moderate intellectual disability [14].

With an increasing number of patients being clinically and molecularly characterized, mutations in the “non-syndromic MCPH genes” are being reported with a broad range of morphologic or functional anomalies such as brain malformations, epilepsy, behavioral problems, abnormal motor development, ataxia, hearing loss, pigment anomalies, dysmorphism, skeletal anomalies, ophthalmological anomalies, cardiac anomalies, etc. [15]. Consequently, the term “MCPH syndromes” was established [4].

For instance, a recent study reports patients harboring *CDK5RAP2* variants manifesting extra-cerebral features such as retinal and cochlear developmental defects and hypothalamic anomalies. Although *CDK5RAP2* is traditionally established as a “non-syndromic MCPH gene”, the presence of further anomalies in these patients is more consistent with the criteria of syndromic microcephaly [16]. It may also be noted that clinical data of *CDK5RAP2* patients are too sparse in most of the reports to exclude extra-neuroglial features in those patients [16]. Therefore, variants of *CDK5RAP2* may also cause multi-organ phenotypes, as suggested by the reports of patients with *CDK5RAP2* variants manifesting CM along with sparse eyebrows [17], congenital cataracts [18], and pigmentary abnormalities [15].

Additionally, a large body of data indicates that the phenotypic consequences of different variants of the same gene can be highly divergent depending on the functional consequences [2,19,20,21,22,23]. For instance, a study has recently grouped *CEP135*-mutated cases of MCPH and MOPD (microcephalic osteodysplastic primordial dwarfism) under the umbrella term of “*CEP135* microcephaly” [2].

These observations have led us to believe that it could be more practical to assign the clinical category of CM (isolated, non-syndromic, or syndromic form of microcephaly) to a specific variant of the relevant gene instead of the gene itself. Furthermore, the causality of a gene with specific phenotypic features should be assigned only when ample and consistent clinical and genetic data are available to draw such a conclusion. Towards this aim, it should be helpful to redefine non-syndromic microcephaly by explicitly specifying the associated phenotypes. Another study reported variants of *ZNF335* (MCPH10, #615095) that cause “severe autosomal recessive primary microcephaly” even though the patients exhibit cardiac, eye, limb, and digital defects, thus consistent with the definition of syndromic MCPH [24]. Such studies can be refined based on the availability of accumulated data to demonstrate that *ZNF335* variants mostly cause multi-organ phenotypes. Otherwise, the observed multi-organ phenotype may be defined as a *ZNF335*-variant-specific phenotype. The same can be said for other genes such as *CEP135* and *PCNT* which cause both non-syndromic and syndromic microcephaly. In those cases where the same variant of a gene causes both non-syndromic and syndromic microcephaly, genetic modifiers or de novo events should also be taken into consideration when assessing the etiology of a multi-organ phenotype [19].

Furthermore, “*ASPM* primary microcephaly” based on biallelic pathogenic variants in *ASPM* is mostly a *typical* non-syndromic MCPH without extra-cephalic abnormalities, yet still manifesting as an intellectual disability [25]. On the contrary, a few studies are showing an additional phenotype of short stature in patients with *ASPM* variants [26,27,28,29,30,31]; thus, short stature may be added to the phenotypic criteria of *ASPM* microcephaly. As a matter of fact, short stature is frequently observed in “non-syndromic MCPH” patients with mutations in *MCPH1*, *WDR62*, *PHC1*, *STIL*, *CDK5RAP2*, and *KNL1* (formerly known as *CASC5*) (Table 1) [32,33,34,35,36,37,38,39,40,41,42,43]. Additionally, the association of short stature is shown in 11/21 (52%) affected members of molecularly and clinically diagnosed “non-syndromic MCPH” patients [2]. Interestingly, patients with variants in *CEP135*, *CENPJ*, and *CEP152* show short stature only when diagnosed with MOPD [2,44]. Apparently, variants in centrosomal genes seem to be more likely to cause multi-organ phenotypes. Nevertheless, establishing a differential, inclusive, and preferably tier-based diagnostic criterion is necessary at this time.

Notably, developmental delay is frequently associated with CM features [45,46,47,48,49], as is evident from the 935 entries of developmental delays in the clinical synopsis of CM in MIM (Mendelian Inheritance in Man) [last checked on 12.09.2022]. Therefore, its inclusion as one of the diagnostic features of CM could be helpful. Similarly, lissencephaly and pachygyria, as well as other architectonic cerebral defects, are frequently observed in patients harboring variants in *ASPM*, *WDR62*, *CEP135*, *MCPH1*, *CIT*, *CENPJ*, and *STIL* and have also been proposed to be included as MCPH-defining features [2]. Less commonly, mild intrauterine growth restriction with postnatal appearance, subnormal motor development, articulation impairment, spasticity, and dysmorphic facial features have also been observed [25]. Hence, it would be of great diagnostic aid if a uniform list of clinical features is established that would guide the clinicians/researchers when defining CM subtypes. Such reports, though greatly valuable for expanding the clinical spectrum of the mutated genes, are not in line with the conventional diagnostic criteria of MCPH.

## 3. Genetic Etiology of Microcephaly Primary Hereditary (MCPH)

Hunting for causal gene variants of microcephaly traces back to 1985, when Pérez Castillo published the first microcephaly-linked locus at cytoband 1q31–1q321 [50]. To date, Online Mendelian Inheritance in Man (OMIM) retrieves 1255 entries (last checked on 6 September 2022) with microcephaly in a clinical synopsis section. According to our knowledge, 32 genes have been reported to cause non-syndromic primary microcephaly (Table 2). Interestingly, the 31st and 32nd genes of MCPH i.e., *SNRPE* and *AKNA*, respectively, are yet to be updated by OMIM (last checked on 16 December 2022) [51,52]. Among these, *ASPM* and *WDR62* are the most frequently mutated genes, accounting for 68% and 14%, respectively, of the reported families [53,54].

The genetic causes of microcephaly provide insights into the basic processes of brain development and neurogenic mitosis [55]. The majority of variants reported in these genes are frameshift or nonsense variants, eventually leading to truncated, absent, or nonfunctional proteins [56]. MCPH proteins have particular roles in mitosis and are expressed in the neuroepithelium during embryonic neurogenesis [4,57]. The mitosis in neural progenitor cells is surpassingly reliant on the optimal function of these proteins. Any aberrations can cause them to prematurely switch from symmetric proliferative to asymmetric differentiating cell division, thus reducing the progenitor pool and consequently the number of neurons as a whole, producing the MCPH phenotype [58].

The proteins encoded by MCPH-associated genes (Table 2) are expressed ubiquitously and most of them (microcephalin, CDK5RAP2, CENP-J, STIL, CEP135, CEP152, CDK6, HsSAS-6, PDCD6IP, and AKNA) are constituent of either centrosome, spindle pole (WDR62, ASPM, and CENP-E) or spindle microtubules (MAP11), during at least part of the cell cycle [52,59,60,61]. Some MCPH-associated proteins are solely localized at the centrosome; others have a nuclear or cytoplasmic localization in interphase, and a centrosomal localization only in mitotic cells [60,62,63]. Due to the centrosomal localization of several MCPH-associated proteins and their role in centriole duplication, these disorders have also been coined as “centriolopathies” [64]. In addition to the centrosome, several MCPH proteins localize at the kinetochore (KNL1, CENP-E, p37, and hBUB1) and midbody (CRIK, KIF14, and MAP11), particularly during the cell cycle, indicating that dysfunctions of kinetochore and midbody also play critical roles in the etiology of primary microcephaly [13,59,65]. Two of the MCPH proteins, kinetochore scaffold 1 (KNL1) and centromere-associated protein E (CENP-E), are essential components of the kinetochore, where the former controls kinetochore formation and proper segregation of chromosomes during mitosis through interaction with MIS12 whereas the latter helps to capture spindle microtubules to the kinetochore [40,66,67]. Intriguingly, *MCPH1*-encoded protein microcephalin is reported to promote telomere replication by interacting with the telomere-binding protein TRF2. This complex facilitates the recruitment of DNA damage response factors to the defective telomeres. Defective telomere replication has thus been implicated as the underlying mechanism of microcephaly [68]. A few MCPH patients have been reported with biallelic loss of function mutations in *CIT* which encodes “citron Rho-interacting kinase (CRIK)” that localizes on the cleavage furrow and midbody of the dividing cells and mediates cytokinesis [69]. After this finding, additional midbody components such as KIF14 and MAP11 were identified as brain size determinants. The components of chromatin-remodeling complexes encoded by *ZNF335* and *PHC1*, the components of the condensin complex encoded by *NCAPD2*, *NCAPH*, and *NCAPD3*, as well as the pre-mRNA-processing spliceosome component encoded by the *SNRPE* gene represent regulators of mitotic division and their dysfunction in early development, are associated with microcephaly [33,51,70,71]. Other than altered cell cycle monitors, mutations in genes encoding the non-cell-cycle regulators such as sodium-dependent lysophosphatidylcholine symporter 1 (NLS1) encoded by *MFSD2A*, ankyrin repeat and LEM domain-containing protein 2 encoded by *ANKLE2*, a subunit of the Golgi coatomer complex (Coatomer subunit beta, COPB2), components of the nuclear lamina encoded by the *LMNB1* and *LMNB2* genes, and the nucleolar protein (ribosomal RNA processing 7 homolog A) encoded by *RRP7A*—which is an essential regulator of ribosomal RNA (rRNA) processing—have also been reported in primary microcephaly patients [53,72,73,74,75].

Taken together, the identified MCPH proteins evidently play critical roles in several cellular processes such as cell division and cell cycle regulation, ciliogenesis, cytokinesis, transcription regulation, kinetochore function, chromosome segregation, chromosome condensation, telomere replication, transmembrane or intracellular transport, lipid metabolism and transportation through the blood–brain barrier, ribosomal RNA processing, pre-mRNA processing, DNA damage response, autophagy, and Wnt signaling (Table 2), implicating a crucial role of all these processes for normal brain development [76,77,78].

**Table 2 cells-12-00642-t002:** List of genes involved in the etiology of MCPH.

Locus	Gene	Protein	Subcellular Localization	Cellular Process	References
** *MCPH1* **	*MCPH1*	Microcephalin	Nucleus (interphase), centrosome (interphase and mitosis)	DNA damage signaling and repair, the regulation of chromosome condensation, cell-cycle progression, telomere replication and repair, and centrosome function	[68,79,80,81]
** *MCPH2* **	*WDR62*	WD-repeat containing protein 62	Centrosome (interphase), spindle poles (mitosis)	Mitogenic kinase signaling, centrosome function, cytoskeletal organization, and cell cycle progression	[63,64]
** *MCPH3* **	*CDK5RAP2*	Cyclin-dependent kinase 5 regulatory subunit-associated protein 2	Centrosome	Centrosome function, DNA damage response, and cell cycle progression	[4,22]
** *MCPH4* **	*KNL1* *	Kinetochore scaffold 1	Kinetochore	Kinetochore assembly, chromosome congression, and mitotic checkpoint signaling	[40,82]
** *MCPH5* **	*ASPM*	Abnormal spindle-like microcephaly-associated protein	Centrosome (interphase), spindle poles (mitosis)	Spindle organization, cytokinesis, centriole biogenesis, and microtubule disassembly	[83,84]
** *MCPH6* **	*CENPJ*	Centromere protein J	Centrosome	Centriole biogenesis, cilium disassembly, and cell cycle progression	[22,85]
** *MCPH7* **	*STIL*	SCL/TAL1-interrupting locus protein	Centrosome	Centriole duplication and cell cycle progression	[4,86]
** *MCPH8* **	*CEP135*	Centrosomal protein of 135 kDa	Centrosome	Centriole biogenesis	[87,88]
** *MCPH9* **	*CEP152*	Centrosomal protein of 152 kDa	Centrosome	Centriole duplication and DNA damage response through ATR-mediated checkpoint signaling	[20,21]
** *MCPH10* **	*ZNF335*	Zinc finger protein 335	Nucleus	Transcription regulation	[4,71]
** *MCPH11* **	*PHC1*	Polyhomeotic-like protein 1	Nucleus	Chromatin remodeling, DNA damage and repair, and cell cycle regulation	[4,33]
** *MCPH12* **	*CDK6*	Cyclin-dependent kinase 6	Cytosol and nucleus (interphase), centrosome (mitosis)	Organization of microtubules, centrosome integrity, and cell proliferation	[60]
** *MCPH13* **	*CENPE*	Centromere-associated protein E	Centromere, kinetochore, spindle midzone	Chromosome alignment and movement toward microtubule bundles	[67,89]
** *MCPH14* **	*SASS6*	Spindle assembly abnormal protein 6 homolog	Centrosome	Centriole duplication	[4,90]
** *MCPH15* **	*MFSD2A*	Sodium-dependent lysophosphatidylcholine symporter 1	Cell membrane	Lipid metabolism and transportation through the blood–brain barrier	[91]
** *MCPH16* **	*ANKLE2*	Ankyrin repeat- and LEM domain-containing protein 2	The endoplasmic reticulum, nuclear envelope	Maintain nuclear envelope morphology, cell division, and proliferation	[92,93]
** *MCPH17* **	*CIT*	Citron Rho-interacting kinase	Cleavage furrow and midbody	Cytokinesis	[94]
** *MCPH18* **	*WDFY3*	WD repeat and FYVE domain-containing protein 3	Cytosol and nucleus	Wnt signaling (autophagy-dependent manner)	[14]
** *MCPH19* **	*COPB2*	Coatomer subunit beta	Golgi apparatus membrane	May regulate autophagy and maintain the integrity of cellular organelles and cell homeostasis	[72]
** *MCPH20* **	*KIF14*	Kinesin-like protein KIF14	Spindle midzone and the midbody	Cytokinesis	[65]
** *MCPH21* **	*NCAPD2*	Condensin complex subunit 1	Cytoplasm, nucleus (interphase), chromatin (mitosis)	Mitotic chromosome condensation	[70]
** *MCPH22* **	*NCAPD3*	Condensin-2 complex subunit D3	Cytoplasm, nucleus (interphase), chromatin (mitosis)	Mitotic chromosome condensation	[70]
** *MCPH23* **	*NCAPH*	Condensin complex subunit 2	Cytoplasm, nucleus (interphase), chromatin (mitosis)	Mitotic chromosome condensation	[70]
** *MCPH24* **	*NUP37*	Nucleoporin Nup37	Nuclear envelop (interphase), kinetochore (mitosis)	Cell cycle progression	[73]
** *MCPH25* **	*MAP11*	Trafficking protein particle complex subunit 14	Spindle microtubules, cleavage furrow and midbody	Cytokinesis and cell abscission	[59]
** *MCPH26* **	*LMNB1*	Lamin-B1	Nuclear lamina	Cell cycle regulation, transcription regulation, and DNA repair	[74,95]
** *MCPH27* **	*LMNB2*	Lamin-B2	Nuclear lamina	Transcription regulation, mitosis, chromosome segregation, and nucleolar morphology	[74,95]
** *MCPH28* **	*RRP7A*	Ribosomal RNA-processing protein 7 homolog A	Nucleolus, centrosomes and cilia	Ribosomal RNA processing, primary cilia resorption, and cell cycle progression	[75]
** *MCPH29* **	*PDCD6IP*	Programmed cell death 6-interacting protein	Nucleus	Cytokinesis, cell proliferation, and abscission and autophagy	[61]
** *MCPH30* **	*BUB1*	BUB1 mitotic checkpoint serine/threonine kinase	Nucleus (interphase), kinetochore (mitosis)	Chromosome congression and spindle assembly checkpoint	[13]
** *MCPH31* **	*SNRPE*	Small nuclear ribonucleoprotein E	Nucleus	Pre-mRNA processing	[51]
** *MCPH32* **	*AKNA*	Microtubule organization protein AKNA	Centrosome	Centrosomal microtubule organization	[52]

Note: Gene and respective encoded protein names are verified from HUGO (Human Genome Organization), the Gene Nomenclature Committee (HGNC) and UniProt Knowledgebase (UniProtKB), respectively. * This gene was previously known as *CASC5.* ATR stands for ataxia-telangiectasia mutated and Rad3-related.

## 4. Model Systems for Microcephaly and Pathomechanisms

The underlying mechanisms of this condition are inferred from the known functions of MCPH proteins investigated during mouse embryonic brain development as well as in cerebral organoids. Knockout mice models of MCPH causative genes (*Mcph1*, *Wdr62*, *Cdk5rap2*, *Aspm*, *Cenpj*, *Znf335*, *Phc1*, *ankle2* (zebrafish), Wdfy3 and *Cit*) mimic the phenotype of reduced brain size discerned in the MCPH patients [33,56,96,97]. Defective DNA damage response (DDR) is also proposed as an underlying mechanism of microcephaly. Ablation of *Mcph1* in mice resulted in elevated genomic instability due to defective homologous recombination repair and mimic microcephaly phenotype. Interestingly, neural progenitor cells (NPCs) examined from these mice were also reported to be vulnerable to ionizing radiation [81]. Similarly, *Knl1*-deficient mice also showed microcephaly as a result of enhanced DNA damage and p53 activation in NPCs [98]. Furthermore, microcephaly observed in the *Cenpj*-deficient mice was the result of elevated levels of DNA damage and apoptosis, thus favoring DDR as a crucial process to control brain size [99]. *Kif14*-knockout mice show small brain size, primarily caused by elevated apoptosis during late neurogenesis in these mice [100]. *Mfsd2a*-knockout mice demonstrate a severe neurodevelopmental phenotype, including microcephaly, anxiety, cognitive defects, and ataxia leading to postnatal lethality [101,102]. Furthermore, condensin II (*Ncaph2*) mutant mice showed significantly reduced brain size, accompanied by chromosome segregation errors, leading to micronucleus formation and increased aneuploidy in daughter cells [70]. In addition, the ablation of *Cenpj*, *Knl1*, *Stil*, *Cep135*, *Cenpe*, and *Copb2* (the orthologs of *MCPH3*, *MCPH4*, *MCPH7*, *MCPH8 MCPH13*, and *MCPH19* loci, respectively) results in the embryonic lethality of mice [56,72,99,103,104]. Conclusively, the viability of MCPH patients mutated for these genes could possibly be explained by the hypomorphic nature of the identified mutations, as compared to more damaging nullimorphic alleles in mice [72,105]. Other than mouse models, microcephaly has also been demonstrated in the cerebral organoids developed from *CDK5RAP2* and *ASPM* mutated patient-derived induced pluripotent stem (iPS) cells, which indicated premature neuronal differentiation as a pathomechanism for the former gene and impaired neuronal functions (due to defective lamination), as well as decreased neural progenitor pool for the later gene [106,107]. Similarly, premature neuronal progenitor differentiation has been proven by yet another study using cerebral organoids developed from iPS cells of Seckel syndrome patients mutated with *CENPJ*, where the authors claim delayed cell cycle entry as a consequence of impaired cilium disassembly [85]. These reports strengthen the conserved and indispensable role of MCPH genes in brain development. Based on the knowledge gained from these models, two different underlying mechanisms—centrosomal and non-centrosomal—emerge for primary microcephaly. In the centrosomal model, it is speculated that the defective MCPH proteins affect microtubule organization at the centrosome in neural progenitor cells, resulting in premature switching from both symmetric proliferative [108] and asymmetric self-renewing to symmetric consumptive division [109]. The cells prematurely lose their proliferative capacity, causing early depletion of the pool of progenitors, and consequently a smaller brain [108,110]. In the non-centrosomal model of microcephaly, the spindle orientation is not altered, yet the depletion of neural progenitor cells can be caused by other mechanisms such as chromosome segregation defects, which lead to aneuploid cells, eventually destined to apoptosis. This leaves too few progenitors to constitute a normal cortical volume, resulting in primary microcephaly [70,111]. In addition to these two models, perturbation in the migration of neurons and glial cells has also been suggested as the cause of primary microcephaly in patients carrying mutations in *WDR62* [112].

## 5. The Shift from Non-Syndromic to Syndromic Microcephaly Remains Unsolved

The non-linear genotype–phenotype relationship in microcephaly patients has been hugely debated in recent years. Most of the MCPH genes are globally expressed but their mutations exceedingly compromise only brain development, a phenomenon that remains only vaguely understood [2]. Below, we discuss some arguments that might explain the etiologic dichotomy of non-syndromic and syndromic microcephaly.

### 5.1. The Fragility of Asymmetric Neuronal Division

During neurogenesis, the formation of a neural tube requires precisely orchestrated temporal and spatial arrangements of cell division [113]. This process is highly sophisticated and demands an organized fashion and proper regulation at every step. As explained above, embryonic neurogenesis conceivably only shows exclusive fragility in this context, and any dysregulation in the balance between symmetric and asymmetric cell division, such as premature switching from symmetric proliferative to asymmetric apical progenitor division [108], and from asymmetric self-renewing to symmetric consumptive apical progenitor division, considerably reduces the number of neural progenitor cells, leaving too few of them to constitute a normal-sized brain [108]. Furthermore, spindle pole formation and centrosome duplication, which serve to maintain polarity to neuronal cells, are also proven to be extremely crucial events in neuronal duplication. Interestingly, centrosomal abnormalities show a less profound influence on extra neurogenic mitosis, as shown by studies of neuroblasts in Drosophila [114]. It is worth mentioning that most of the centrosomal proteins function together; MCPH1 and WDR62 are physically interacting partners and make a complex with Cep63 as well, and the knockdown of *CEP63* abolishes MCPH1-WDR62 interaction [64]. In addition, Cep63, ASPM, and WDR62 recruit CENP-J (also known as CPAP) to carry out centriole biogenesis [64]. If we trace expression patterns, it is seen that all of them are highly expressed in the cerebral cortex, localized in close intracellular vicinity of each other (centrosome/spindle pole), and are interconnected, indicating that they mediate each other’s function. Not only do these make complexes but they also exhibit functional dependency, which collectively supports the idea of the sensitivity of asymmetric division towards the dysfunctioning of either of these proteins. Hence, it is apparent that neuronal cells are specifically sensitive to centrosomal defects, leading to depletion of the neural progenitor pool.

### 5.2. Role of Genetic Modifiers/Multiple Allelism

Microcephaly has long been classified as a pure monogenic yet heterogeneous disorder; however, recent pieces of evidence for an oligogenic model of microcephaly are also emerging. The importance of genetic modifiers in phenotypic variability has attained special attention in recent years. A genetic modifier can add to the phenotypic variability due to functional co-dependencies among the genes of related pathways or genetic interaction by enhancing or alleviating the severity of the disease [115]. Due to the reduced cost of genomic sequencing, it is now increasingly feasible to hunt for additional pathogenic variants. The presence of more than one pathogenic variant can completely shift the phenotypic paradigm, as has been shown in a recent study that elegantly proves this fact on functional grounds. In a multiplex Pakistani family, the severity of phenotypes was noted in patients carrying the heterozygous *PCNT* variant (modifier), additionally harboring the homozygous variant of *CENPJ*, as compared to those cases where *PCNT* mutation was not segregated [19]; one loop manifested in Seckel syndrome caused by a pathogenic *CENPJ* variant (NM_018451.4:c.3586G > A; p.(Asp1196Asn)) which was also segregating in the second loop. However, the second loop manifested microcephalic osteodysplastic primordial dwarfism type II (MOPDII)—a phenotypically severe condition—due to the presence of an additional *PCNT* variant (NM_006031.5;c.5767C > T; p.(Arg1923*)). Further, this study also provided evidence that heterozygous missense variants of *WDR62*, *CEP63*, and *RAD50* worsen the MCPH phenotype in patients harboring a homozygous *ASPM* mutation [19]. Another study also reported the severity of the MCPH phenotype caused by genetic modifiers. In this report a homozygous stop mutation, c.1605dupT; p.(Glu536*) of *WDR62*, caused MCPH in one patient, whereas the other affected members presented severe phenotypic manifestations in the presence of a chromosomal duplication of 17q25-qter on one allele and a missense mutation, c.3361T > G; p.(Phe1121Val) of *TBCD*, on a second non-duplicated allele [116]. Conclusively, the authors proposed that the modifying effect of *TBCD* resulted in the severity of the clinical manifestations in the patients. Interestingly, digenic tri-allelic inheritance in MCPH patients, with the combination of genes *WDR62*/*CDK5RAP2*, *ASPM*/*WDR62*, and *WDR62*/*CEP135* has also been reported. However, the authors did not show the modifying effects of additional variants on the clinical manifestations of the patients. It should be noted that they did not observe any modifying effects of *wdr62* or *aspm* ablations in zebrafish upon knockout of *knl1*. However, the quadri-allelic ablation of *wdr62* and *aspm* demonstrated primary microcephaly in zebrafish [117].

In this context, the role of modifiers in exacerbating the phenotype has also been reported in animal models. A previous study analyzed similar modifier effects for *Gpr63* in mice, where *Ttc21b* was ablated in homozygous form; *Ttc21b* is already known to control brain size, and mutations in this gene cause ciliopathies, including microcephaly [118]. A CRISPR/Cas9-generated heterozygous mutant *Gpr63* allele in *Ttc21b* null showed an increased incidence of neural tube defects, which was not observed in *Ttc21b* homozygous mutants. Intriguingly, a further severity of phenotypes (early embryonic lethality, neural tube defects, exencephaly, polydactyly of forelimbs and hindlimbs, and spina bifida aperta) was observed in *Gpr63* and *Ttc21b* double homozygous mutants, which were not shown in the single mutants [118].

Additionally, genes encoding centrosomal proteins such as *ASPM* and *WDR62* when mutated cause centrosomal defects leading primarily to the MCPH phenotype. Interestingly, tri-allelism (knockout of two mutant copies of one gene and only one mutant copy of another gene), *Wdr62* (+/−), and *Aspm* (−/−), in mice embryos show comparatively severe cellular defects, and thus microcephaly, showing predominant effects on late neurogenesis. A similar effect was noticed in the case of *Wdr62* (−/−) and *Aspm* (+/−), suggesting severe mutational effects. However, in the case of quadriallelism, the double-knockout mice of *Wdr62* (−/−) and *Aspm* (−/−) showed early embryonic lethality [64]. A similar concept was supported by another study where multiple allelism was analyzed in zebrafish; a double-knockout of genes encoding centrosomal proteins such as *Aspm* and *Wdr62* and one non-centrosomal *Knl1*. The latter demonstrated no exacerbation of the phenotype. This shows that oligogenic inheritance may show a variable phenotype when the genetic variants are functionally relevant and/or have genetic interaction in one way or another [117].

### 5.3. Altered Splicing Events

The concept of tissue-specific splicing factors, and hence tissue-specific splicing, is a very well-established phenomenon and can possibly contribute to explaining brain-specific phenotype [119]. It is indicated in a recent study where MCPH-protein KNL1 has been investigated in cerebroids and neural progenitor cells generated by reprogramming the human embryonic stem cells (hESCs) harboring a CRISPR/Cas9-mediated knock-in missense variant (c.6125G > A; p.Met2041Ile) of *KNL1* [82]. They observed that the mutation inhibited an exon splicing enhancer (ESE) site and generated an exon splicing silencer (ESS) site. This newly generated ESS site is recognized by the heterogeneous nuclear ribonucleoprotein A1 (HNRNPA1) which acts as an inhibitory splicing factor and thus lowers the expression of KNL1. Authors revealed that HNRNPA1 is dominantly expressed in the brain and only mildly expressed in human fibroblasts and neural crest cells. Therefore, it affects the expression of KNL1 in the brain only, but not in primary fibroblast and neural crest cells. The absence of KNL1, a crucial mediator of chromosome alignments during cell division, results in aneuploidy, leading to reduced cell growth, apoptosis, and reduced size of the cerebroids. However, the patient-derived cells (primary fibroblasts and lymphoblastoid cells), as well as primary fibroblasts and neural crest cells differentiated from isogenic mutant hESC cells, did not show any aberrant cellular phenotype [40,82]. As the mutational effects were observed only in the neural progenitors, this study concluded that the *KNL1* mutation exclusively affects neurogenesis, thus causing a brain-restricted phenotype.

*KNL1* is not the only gene that has a splice site variant; MCPH genes such as *MCPH1*, *WDR62*, *CDK5RAP2*, *ASPM*, *CEP135*, *ZNF335*, *SASS6*, *MFSD2A*, *CIT*, *KIF14*, *NCAPD2*, *NCAPD3*, and *LMNB1* with splice site variants were also reported in MCPH patients [15,63,69,70,71,86,102,120,121,122,123,124]. These splice variants may act in the same manners as observed in the case of KNL1 splice variants and may explain the phenotype of isolated microcephaly. Furthermore, a recent study shows *SNRPE* variants causing congenital non-syndromic microcephaly; this gene encodes SmE proteins involved in the assembly of the spliceosome complex. The mutation impaired the assembly of the spliceosome components, which leads to the global dysregulation of mRNA splicing in patient-derived human primary fibroblasts and SmE-depleted zebrafish, as well as HEK293 cell line [51]. The RNA profiling of morpholino-injected zebrafish (head and tail only) showed the clustering of retained introns in several transcripts due to SmE deficiency. Notably, authors observed a distinguishable expression pattern of transcripts of head and tail regions with the majority of the downregulated genes enriched for the development of the central nervous system. These data suggest that neurogenesis/brain development is exclusively vulnerable to the splicing alterations and cause a brain-specific phenotype, as the patient of this study presented a non-syndromic form of microcephaly only [51].

### 5.4. Hypomorphic and Loss-of-Function Variants

It has been indicated that the mutations causing complete loss of the MCPH protein may have severe, multi-organ manifestations (syndromic microcephaly), whereas mutations partially compromising the function of the protein may cause a brain-restricted phenotype [125]. This argument is elegantly supported by establishing the genotype–phenotype correlation of different components of the nuclear pore complex, p37, NUP107, and NUP133, which showed phenotypic variability in microcephalics; it was intriguing to observe that patients segregating a hypomorphic *NUP37* mutation did not show any symptoms of steroid-resistant nephrotic syndrome (SRNS), which was diagnosed only in the individuals segregating the *NUP107* loss-of-function variant [73]. Authors modeled both hypomorphic and null mutations of *NUP107* in zebrafish by CRISPR/Cas9-mediated genome editing and showed that truncating mutation (c.50_56del7; p.Thr81Argfs*74) causes early lethality and developmental defects including microcephaly, whereas in-frame hypomorphic mutation (c.137_139del; p.Ala46del) was compatible with embryonic survival and did not cause any obvious phenotype [73]. These findings thus reflect the inverse relation of a residual function of the gene with the general severity of the disease, broadening the phenotypic spectra [53].

Additionally, the loss-of-function variant in NUP133—the direct binding partner of NUP107—leads to Galloway–Mowat syndrome, which is characterized by microcephaly associated with early onset nephrotic syndrome and hiatus hernia. The mutation causes an abrogation of interaction between NUP107 and NUP133, leading to a reduction in both proteins, which can be related to the severity of the disease [126]. Thus, it is convincing to deduce that hypomorphic mutations show milder phenotypes, whereas loss-of-function mutations lead to severe medical consequences affecting brain development along with other organs such as kidneys in SRNS. Intriguingly, hypomorphic variants of *CENPJ* and *CEP152* are reported to cause only a brain-restricted phenotype (non-syndromic microcephaly), whereas complete loss-of-function variants of these genes result in Seckel syndrome [21,127].

### 5.5. De Novo Autosomal Dominant Mutations

De novo variants are also reported to cause primary microcephaly. A study presented seven individuals who were reported with de novo variants in *LMNB1*, all manifesting primary microcephaly associated with additional features such as relatively short stature, neurogenic scoliosis, and severe feeding difficulties. Two of these individuals segregating splice variants causing exon elongation also demonstrated hypotonia, spastic tetraparesis, and gingival hypertrophy. The head circumferences ranged from −3.6 to −12 SD, showing the severity of the disease [123]. Similarly, de novo mutations in *KAT6A*, *TUBB5*, *ASXL3*, *SPOCK1*, *CTNNB1*, and *KIF11* were also observed to cause severe syndromic microcephaly [128,129,130,131,132,133]. A recent Korean study presented a cohort of 34 patients with primary microcephaly and 6 patients with postnatal microcephaly. Authors identified de novo mutations in 10 genes (*GNB1*, *GNAO1*, *TCF4*, *ASXL1*, *SMC1A*, *KMT2A*, *VPS13B*, *ACTG1*, *EP300*, and *KMT2D*) segregating in the microcephalics with a broad phenotypic spectrum, including developmental delay and infantile hypotonia consistent in all, variably associated with a movement disorder (in 7 patients) and other anomalies (congenital heart defect, cleft lip/palate, and skeletal anomaly) in 17 patients [48]. Furthermore, another study showed that a de novo heterozygous synonymous variant of *EFTUD2*, which encodes a spliceosomal component, caused postnatal microcephaly associated with mandibulofacial dysostosis (hearing loss, epileptic seizures, micro retrognathism, malar hypoplasia, and dental anomalies) [134]. Taken together, these studies indicate that the severity of disease in CM cases is dependent on the nature of the mutation, and thus the syndromic cases should be explored for the de novo variants as the first line of strategy. However, the possibility of the de novo variants could not be ruled out, particularly in those cases carrying homozygous/compound heterozygous variants manifesting severe phenotypes.

## 6. Conclusions

In this review article, we have highlighted the inconsistencies in the differential diagnosis of congenital microcephaly (CM) and outlined the accumulating evidence that should be critically considered when evaluating and defining CM subtypes. Keeping in mind the limited understanding of the molecular mechanism of the paradigm shift from non-syndromic to syndromic microcephaly, we have documented the molecular causes necessary for the shift from brain-restricted to extra-neurological features in CM cases. In conclusion, we recommend that genetic modifiers, de novo variants, and brain-specific splicing events be considered when evaluating the spectrum from brain-specific to multi-organ phenotypes associated with particular genes.

## Figures and Tables

**Table 1 cells-12-00642-t001:** MCPH cases reported with short stature.

Gene	Variant	No. of Patients	Additional Phenotypes	Diagnosis/Common Features	Study
** *ASPM* **	c.688delG/c.3340delA	2	SS	MCPH/SH, SGP, ID, BP	[26]
	c.688delG/c.9190C > T	1	SS		
	c.3979C > T	1	SS		
	c.9541C > T	1	SS		
**Not identified**	-	1	SS, syndactyly, strabismus	MCPH/SH, ID	[27]
** *ASPM* **	c.6854_6855del	5	SS	MCPH/ SH, ID, DD, speech delay	[28]
** *ASPM* **	c.3742-1G > C	2	SS	MCPH/SGP, ID, epilepsy, speech delay	[29]
** *ASPM* **	c.9286C > T	3	SS, hearing impairment	MCPH, ID	[30]
	c.3055C > T	8	SS		
	c.9319C > T	3	SS		
**Not identified**	-	2	SS	MCPH	[30]
		2	SS, joint deformity		
		3	SS, cataract		
		3	SS, strabismus, ataxia		
		2	SS		
** *ASPM* **	c.3945_3946delAG/c.8191_8194delGAAA	1	SS	MCPH, ID	[31]
** *PHC1* **	c.2974C > T	2	SS	PM/SH	[33]
** *STIL* **	c.2150G > A	2	SS, sleep defects	MCPH, ID	[32]
** *STIL* **	c.4849C > T	2	SS, Holoprosencephaly	MCPH, ID	[27]
** *MCPH1* **	150–200 kb deletion of first 6 exons	6	SS	MCPH, ID	[34]
** *MCPH1* **	61 kb homozygous deletion of first 8 exon	1	SS, strabismus, ventriculomegaly	CM, ID, SGP, speech delay	[35]
** *WDR62* **	c.668T > C	2	SS, Cortical malformations: hemispherical asymmetry, diffuse pachygyria, thick gray matter, indistinct gray-white matter junction, corpus callosum and white matter hypoplasia	CM, ID, DD	[36]
** *WDR62* **	c.1027C > T	2	SS, delayed motor skills	MCPH, ID, DD, speech delay	[37]
** *WDR62* **	c.1598A > G	2	SS, structural abnormalities and cortical malformation of the brain, motor impairment, increased deep tendon reflexes, flexible joint contractures, sensorineural hearing loss, vertical nystagmus, focal seizures	MCPH, ID, DD, speech delay	[38]
** *WDR62* **	c.2195C > T; p.(Thr732Ile)	5	SS, seizures	MCPH, ID, speech delay	[43]
** *CDK5RAP2* **	c. 4441C > T	1	SS at birth (only), motor delay Simian crease, large map-like hyperpigmentation, tic disorder	MCPH, SPG, ID, DD, speech delay	[41]
** *CDK5RAP2* **	c.524_528del/c.4005-1G > A	1	SS, delayed bone age, asthma, sleep apnea	MCPH, ID, DD, speech defects	[42]
** *CDK5RAP2* **	c.448C > T; p.(Arg150*)	3	SS	MCPH, DD, ID, speech delay	[43]
** *KNL1* **	c.6125G > A	4	SS	MCPH, ID, DD, speech delay	[40]
** *KNL1* **	c.6125G > A	4	SS, cerebellar vermis hypoplasia	MCPH, ID, DD, speech delay	[39]

Note: PM: primary microcephaly, SH: small head, SS: short stature, SPG: simplified gyral pattern, ID: intellectual disability, BP: behavioral problems, and DD: developmental delay.

## Data Availability

Not valid.

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
