# Peer review of "Congenital Microcephaly: A Debate on Diagnostic Challenges and Etiological Paradigm of the Shift from Isolated/Non-Syndromic to Syndromic Microcephaly"

_cells, 2023, doi:10.3390/cells12040642_

Round 1
Reviewer 1 Report
In this review, authors addresses the inconsistencies in the field in defining the congenital microcephaly. The authors put forward the argument to the need of accessing genetic modifiers, de novo variant, splicing events and hypomorphic and loss of function variants when evaluating CM and its subtypes. The authors did a thorough job in introducing the disease and addressing the inconsistencies. Section 5 is well written in addressing the possible mechanisms that causes the contrast between non-syndromic and syndromic microcephaly.
However, I feel that section 4 (Model systems for microcephaly and pathomechanisms) need to be expanded more. DNA damage response and telomere dysfunction as the underlying mechanism, for example, are not addressed in the section. The section is missing some references that I believe should be part of the paper
For example:
Zhou ZW, Tapias A, Bruhn C, Gruber R, Sukchev M, Wang ZQ. DNA damage response in microcephaly development of MCPH1 mouse model. DNA Repair (Amst). 2013 Aug;12(8):645-55. doi: 10.1016/j.dnarep.2013.04.017. Epub 2013 May 15. PMID: 23683352.
Cicconi A, Rai R, Xiong X, Broton C, Al-Hiyasat A, Hu C, Dong S, Sun W, Garbarino J, Bindra RS, Schildkraut C, Chen Y, Chang S. Microcephalin 1/BRIT1-TRF2 interaction promotes telomere replication and repair, linking telomere dysfunction to primary microcephaly. Nat Commun. 2020 Nov 17;11(1):5861. doi: 10.1038/s41467-020-19674-0. PMID: 33203878; PMCID: PMC7672075.
Reviewer 2 Report
The authors reviewed the genetic factors of CM and highlighted the phenotypic variabilities from a brain-restricted phenotype to a muti-organ phenotype. They summarized that brain-specific splicing event, organ-restricted protein expression, genetic modifiers, autosomal dominant de novo variants may contribute to manifest shifts. Although they aimed to expand understanding of heterogeneity, there are still some points to be added.
1. The authors mentioned the genes related with MCPH in Part 3 “Genetic etiology of congenital microcephaly”. It is better to add the variants in genes with established disease phenotypes in humans other than MCPH or use the term of “MCPH” instead of “CM” in the title and main body.
2. Based on the latest requirements of our journal, the Abstract should be a total of 200 words maximum and should follow the style of structured abstracts: Background, Methods, Results, and Conclusion. We hope the authors could refine the Abstract to increase readability:
3. Information in Table 2 is very limited. It is better to add information such as the expression location (centrosome, spindle pole or spindle microtubules) and involved cellular processes. Meanwhile, the authors could simplify the related context for better reading.
4. The authors declared “a large body of data” (line 107) or “recent pieces of evidence” (line 294), while the citations are missing. Please check through the manuscript.
5. The authors have listed most genes involved in the etiology of MCPH, but the information is still vague. To instruct clinical practice better, we hope the authors could add a Discussion part to conclude if there exist certain genes or variants implicating certain subtypes (e.g., genes involved in certain cell process, splicing events or nonsense mutations, etc.)?
